# Synergized regulation of NK cell education by NKG2A and specific Ly49 family members

Xiaoqian Zhang [1,2], Jin Feng [1,2], Shasha Chen[1], Haoyan Yang[1] & Zhongjun Dong[1]*

Mice lacking MHC class-I (MHC-I) display severe defects in natural killer (NK) cell functional maturation, a process designated as "education". Whether self-MHC-I specific Ly49 family receptors and NKG2A, which are closely linked within the NK gene complex (NKC) locus, are essential for NK cell education is still unclear. Here we show, using CRISPR/Cas9-mediated gene deletion, that mice lacking all members of the Ly49 family exhibit a moderate defect in NK cell activity, while mice lacking only two inhibitory Ly49 members, Ly49C and Ly49I, have comparable phenotypes. Furthermore, the deficiency of NKG2A, which recognizes non-classical MHC-Ib molecules, mildly impairs NK cell function. Notably, the combined deletion of NKG2A and the Ly49 family severely compromises the ability of NK cells to mediate "missing-self" and "induced-self" recognition. Therefore, our data provide genetic evidence supporting that NKG2A and the inhibitory members of Ly49 family receptors synergize to regulate NK cell education.

[1] School of Medicine and Institute for Immunology, Beijing Key Lab for Immunological Research on Chronic Diseases, Tsinghua University, Beijing 100084, China. [3] These authors contributed equally: Xiaoqian Zhang, Jin Feng. *email: dongzj@mail.tsinghua.edu.cn

Natural killer (NK) cells play an important role in the rejection of "unwanted" cellular targets, such as tumours, virally infected cells and allogeneic bone marrow[1–4]. NK cells distinguish normal self-cells from abnormal non-self-cells mainly by surveying the expression of major histocompatibility complex class I (MHC-I) molecules. MHC-I molecules on target cells prevent NK cells from self-destruction by engaging NK cell inhibitory receptors[5–8]. Thus, a lack of MHC-I on target cells will lead to NK-cell activation, a process termed missing-self recognition[9–11]. Therefore, MHC-dependent inhibitory signal is critical for NK-cell tolerance during the effector phase. However, NK cells in MHC-I-deficient mice are functionally hypo-responsive, indicating that MHC-I molecules are compulsorily required for the acquisition of NK-cell functions, a process designated as NK-cell licensing or education[12–15].

MHC-I molecules are further divided into classical MHC-Ia and non-classical MHC-Ib, both of which associate with β2 microglobulin (β2M)[16]. As a result, the deficiency of β2 M will lead to the loss of both types of MHC-I molecules. NK-cell receptors that specifically bind MHC-Ia include killer-cell immunoglobulin-like (KIR) family receptors in humans and the Ly49 family of inhibitory receptors in mouse, while MHC-Ib, such as HLA-E in humans and Qa1 in mice, is recognized by NKG2 and invariant CD94 heterodimers in both species[17–20].

Mouse Ly49-family receptors are typical type II C-type lectin-like transmembrane receptors that include both inhibitory and activating receptors characterized by the presence or absence of immune-receptor tyrosine-based inhibitory motif (ITIM) domains in the cytoplasmic tail[21–24]. Similarly, NKG2A and its different splice variants NKG2B, are inhibitory receptors that contain two ITIMs in their cytoplasmic tail[25–27]. During NK-cell effector processes, MHC-I engagement of these NK-cell inhibitory receptors induces the phosphorylation of tyrosine residue in the ITIM domain, which further recruits the Src-homology 2 (SH2) domain-containing protein tyrosine phosphatases, such as SHP-1, to transduce inhibitory signals[24,28]. Although SHP-1 activation is considered as a negative signal for NK-cell activation, its deletion largely compromises NK-cell activity[28,29], suggesting that this phosphatase is required for MHC-I-regulated NK-cell education. The expression of MHC-I-specific receptors strongly correlates with NK-cell activity[15,30–32]. The most comprehensive genetic study of this family has revealed that downregulation of these receptors reduces NK-cell-mediated immuno-surveillance[33]. Nevertheless, convincing genetic evidence that completely support the hypothesis that MHC-I molecules regulate NK-cell education via their engagement with the Ly49 family and/or NKG2A is still lacking, largely because the Ly49-family members highly share sequence similarity and are closely positioned together within the NK-cell gene complex (NKC) on chromosome 6.

Using CRISPR/Cas9-mediated genome editing, we reveal that inhibitory members of Ly49 family are critical for NK-cell education, while NKG2A provides a synergistic effect with Ly49 family. This finding will help to understand the mechanism underpinning NK-cell tolerance.

## Results

**Generation of mice lacking all members of Ly49 family**. The expression of MHC-I-specific receptors strongly correlates with NK-cell activity[15,30–32]. However, these results are not sufficient to support the hypothesis that MHC-I molecules regulate NK-cell education by engaging inhibitory receptors expressed from the NKC locus. We initially decided to generate mice lacking the entire NKC locus to address this issue. Due to the limitations of the CRSIPR/Cas9 system, we failed to obtain the targeted animals,

even after several attempts. We then tried to obtain mice with a 1.4 Mb deletion of the locus encoding the Ly49 family. As shown in Fig. 1a, nine inhibitory receptors and two activating receptors in the Ly49 family are located within the NKC locus in C57BL/6 mice. We designed one pair of guide RNAs that specifically targets two sites, *Klra2* (Ly49B) and *Klra17* (Ly49Q)[22]. These gRNAs were co-injected into pure C57BL/6 fertilized eggs together with the enzyme Cas9. By performing genomic PCR to screen the genes between *Klra2* to *Klra17*, we successfully obtained three mutant lines lacking all Ly49-family receptors among 45 pups (hereafter referred to as Ly49s KO mice) (Supplementary Fig. 1a). In addition, genome sequencing further confirmed that the Ly49 family-deficient mice carried a deletion of a large fragment, 1,415,735 base pairs, starting from *Klra2* and extending to *Klra17* (Supplementary Fig. 1b). The mRNAs of all Ly49-family genes were not detectable in IL-2-expanded NK cells isolated from these mutants (Supplementary Fig. 1c). Because *Klra17* is mainly expressed on plasmacytoid dendritic cell (pDC) and macrophage, but not NK cell[34], we further confirmed that *Klra17* was detectable in the splenocytes from wild-type (WT) but not Ly49s KO mice (Supplementary Fig. 1c). Using flow cytometry, we finally validated that the mutant mice completely lacked the Ly49-family receptors (Fig. 1b).

**Ly49 family is dispensable for NK-cell development**. NK cells undergo functional and developmental maturation along with sequential expression of multiple Ly49-family receptors[35]. We thus explored whether the Ly49-family deficiency might affect NK-cell development and differentiation. Compared with the wild-type littermates, the percentages and absolute numbers of splenic NK cells were slightly but significantly increased in Ly49s KO mice (Fig. 1c, d). However, we did not notice any developmental advantage of the Ly49s KO mice, as evidenced by the relative percentages of four NK-cell subsets based on the distinct expression of two markers, CD11b and CD27 (Fig. 1e, f).

KLRG1 expression is downregulated in MHC-I-deficient mice[36]. Consistently, we found that the frequency of KLRG1+ NK cells was significantly decreased in Ly49-deficient NK cells (Fig. 1g). We further analysed the expression profile of other NK-cell receptors, particularly receptors located within the NKC locus, and found that several receptors in Ly49s KO mice were marginally downregulated, such as CD117, NKG2D, DNAM-1 and NKG2A. However, the intensity of NKR-P1C (also known as NK1.1) was considerably increased in Ly49s KO mice (Fig. 1g).

Because Ly49-family receptors did not display a restricted expression pattern in cells of the haematopoietic lineage[37], we performed bone marrow (BM) chimeric assay in which Ly49 family deficient bone marrow cells were transferred to immuno-deficient $Rag1^{-/-}\gamma c^{-/-}$ mice. We did not perceive the increased percentages and absolute numbers or the altered differentiation of Ly49 family deficient NK cells (Fig. 1h–j), suggesting that the Ly49 deletion extrinsically affects the pool of NK cells. In addition, Ly49s KO mice had comparable numbers of T cell, pDC, conventional DC (cDC), neutrophil and macrophage in the spleen and BM (Supplementary Fig. 1d). Type-I innate lymphoid cells (ILC1s) were normally detected in the liver of Ly49s KO mice (Supplementary Fig. 1e).

**Ly49-family deficiency moderately impairs NK-cell activity**. We then explore the role of the Ly49 family in NK-cell responsiveness. Resting splenic NK cells were first co-incubated with RMA-S and YAC-1 cells, which are representative targets that trigger "missing-self" and "induced-self" responses, respectively. The Ly49-deficient NK cells exhibited a significant reduction in IFN-γ production and the expression of CD107a, a marker of NK-cell

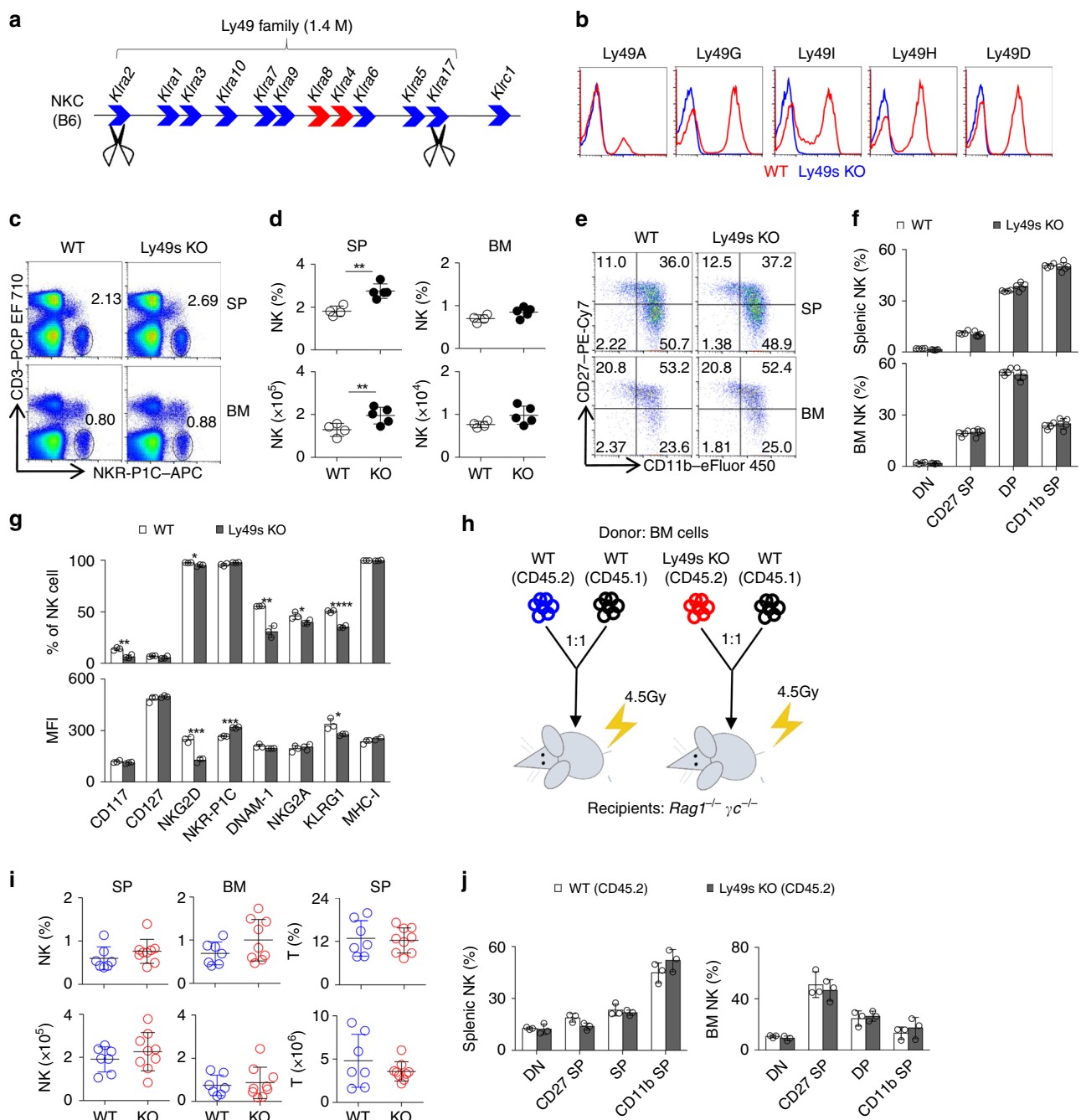

**Fig. 1** Ly49 family is dispensable for NK-cell development. **a** Diagram of Ly49-family genes in the NKC locus. Blue filled arrows denote inhibitory receptors and red filled arrows denote activating receptors. Scissors represent CRISPR gRNAs. **b** Flow cytometry analysis of the expression of Ly49-family receptors on splenic NK cells (gated CD3−NKR-P1C+) from WT (red line) and Ly49s KO (blue line) mice. **c, d** Representative flow cytometry plots (**c**) and quantification (**d**) of NK cells (gated CD3−NKR-P1C+) in the spleen (SP) and bone marrow (BM) of WT and Ly49s KO mice. **e, f** Representative flow cytometry plots (**e**) and percentages (**f**) of gated CD3−NKR-P1C+ NK cells in the four stages of development, including DN (CD27−CD11b−), CD27 SP (CD27+CD11b−), DP (CD27+CD11b+) and CD11b SP (CD27−CD11b+), in the spleen and BM from WT and Ly49s KO mice. **g** The percentage and mean fluorescence index (MFI) of the indicated molecules in gated splenic CD3−NKR-P1C+ NK cells except NKR-P1C and NKp46 in gated splenic CD3−CD122+ NK precursor cells from WT and Ly49s KO mice. **h** Experimental design of bone marrow chimera assay. **i** Quantification of NK cells (gated CD45.2+CD3−NKR-P1C+) and T cells (CD45.2+CD3+NKR-P1C−) in the spleen and BM from chimeric recipient mice (7–9 mice pooled from two independent experiments). **j** Percentages of four NK-cell subsets (gated CD45.2+CD3−NKR-P1C+) NK cells in the spleen and BM from chimeric recipient mice. Each symbol represents an individual mouse. Data shown represent two (**j**) or at least three (**c**–**g**) independent experiments. Mean ± SD is shown. *$p < 0.05$, **$p < 0.01$, ***$p < 0.001$ and ****$p < 0.0001$. Unpaired Student's *t*-tests (two-tailed) was used to calculate these values. Source data are provided as a Source Data file

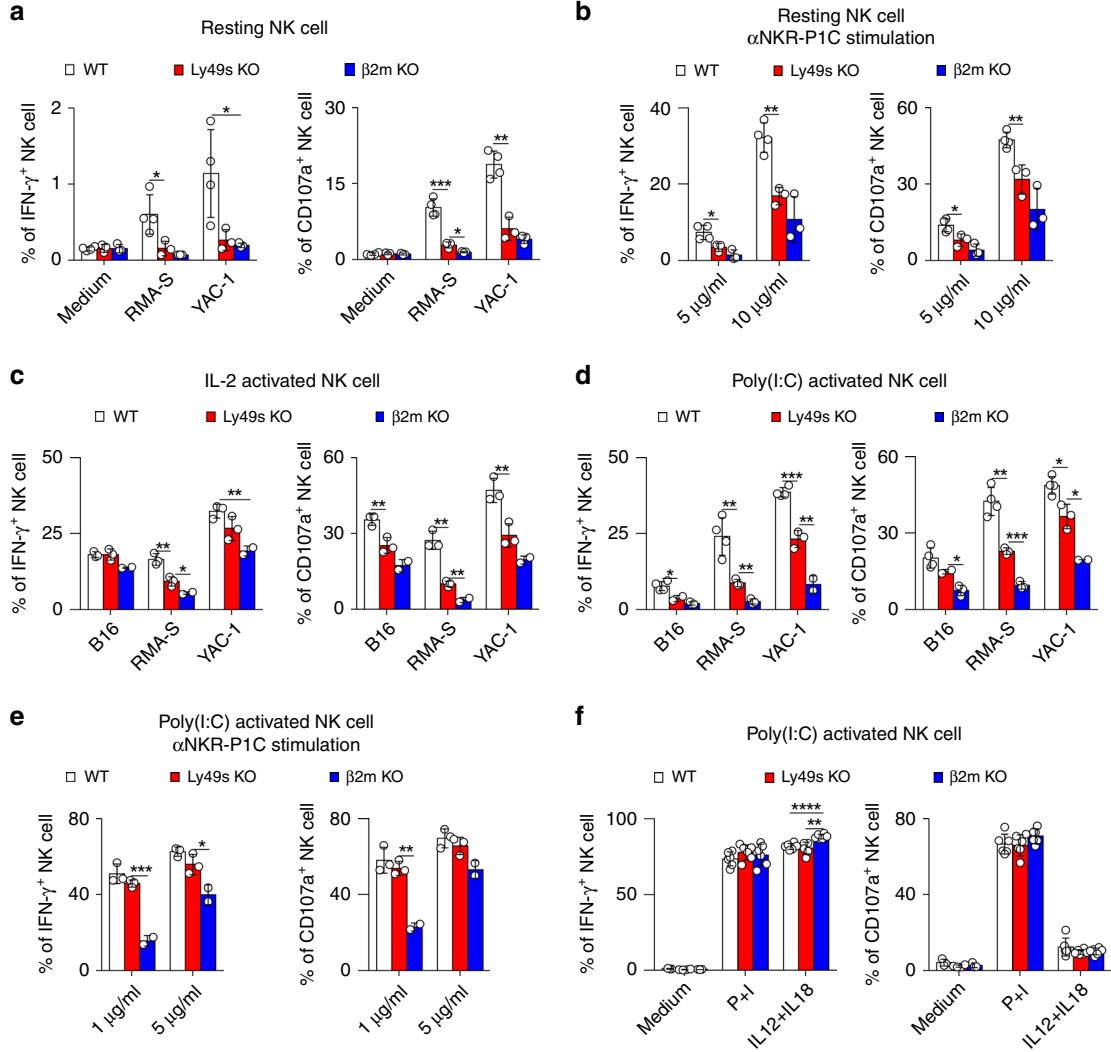

**Fig. 2** Ly49 family-deficient mice display moderate defect in NK-cell activity in vitro. **a** Resting splenocytes from the indicated mice were stimulated with tumour target cells and medium served as the negative control. Percentages of IFN-γ⁺ (left panel) or CD107a⁺ (right panel) gated CD3⁻NKp46⁺ NK cells were analysed. **b** Similar to (**a**), but cells were stimulated with different concentrations of plate-bound antibody against NKR-P1C (clone PK136). **c** Similar to (**a**), but NK cells were pre-cultured with IL-2 for 5 days. **d–f** Poly(I:C)-primed splenocytes from the indicated mice were stimulated with the indicated stimuli, tumour target cells (**d**), plate-bound anti-NKR-P1C antibody (**e**) and phorbol-12-myristate-13-acetate (PMA) plus ionomycin (P + I) and IL12 plus IL18 (**f**). Medium served as the negative control. Percentages of IFN-γ⁺ (left panel) and CD107a⁺ (right panel) gated CD3⁻NKp46⁺ NK cells were analysed. Each symbol represents an individual mouse. Data shown represent two (**c**, **f**) or at least three (**a**, **b**, **d**, **e**) independent experiments. Mean ± SD is shown. *$p < 0.05$, **$p < 0.01$, ***$p < 0.001$ and ****$p < 0.0001$. Unpaired Student's $t$-tests (two-tailed) was used to calculate these values. Source data are provided as a Source Data file

degranulation, in response to both stimuli (Fig. 2a). Resting NK cells were stimulated with a plate-bound antibody against NKR-P1C to test whether the activating receptors in Ly49-deficient NK cells were functionally licensed. Consistently, Ly49-deficient NK cells showed a reduced response to the crosslinking of NKR-P1C (Fig. 2b). We also observed this defect in bone marrow chimaeras, suggesting that the Ly49-family deficiency intrinsically impaired NK-cell function (Supplementary Fig. 2a).

We then conducted other in vitro assays using different sources of NK cells to confirm our observations. Both IL-2-activated and poly(I:C)-activated NK cells isolated from Ly49s KO mice displayed moderate defects in IFN-γ production and CD107a release upon stimulation with haematopoietic RMA-S and YAC-1 cells and non-haematopoietic B16 cells (Fig. 2c, d). In contrast to resting NK cells, poly(I:C)-activated NK cells from Ly49s KO mice showed a less defect in the response to NKR-P1C crosslinking (Fig. 2e), likely suggesting that the defective NK-

cell activity might be bypassed by poly(I:C)-induced cytokines. A subsequent experiment in which Ly49-deficient NK cells displayed an intact competence towards IL-12 plus IL-18 stimulation supported this hypothesis. Finally, the defect in Ly49-deficient NK cells was not due to potential issues with the signalling machinery, as we observed a normal response of Ly49-deficient NK cells to polyclonal stimuli, phorbol-12-myristate-13-acetate (PMA) plus ionomycin (Fig. 2f).

Three in vivo assays were performed to further assess the defective NK-cell activity in Ly49s KO mice after poly(I:C) treatment. First, β2M-deficient splenocytes were chosen as target cells representing the "missing-self" response mediated by NK cells[38]. As expected, the Ly49-family deficiency significantly impaired NK-cell-mediated rejection of MHC-I-deficient haematopoietic cells (Fig. 3a). We also obtained a similar result when RMA-S cells were used as targets in a peritoneal clearance assay (Fig. 3b). Thus, Ly49s KO mice exhibited defective responses

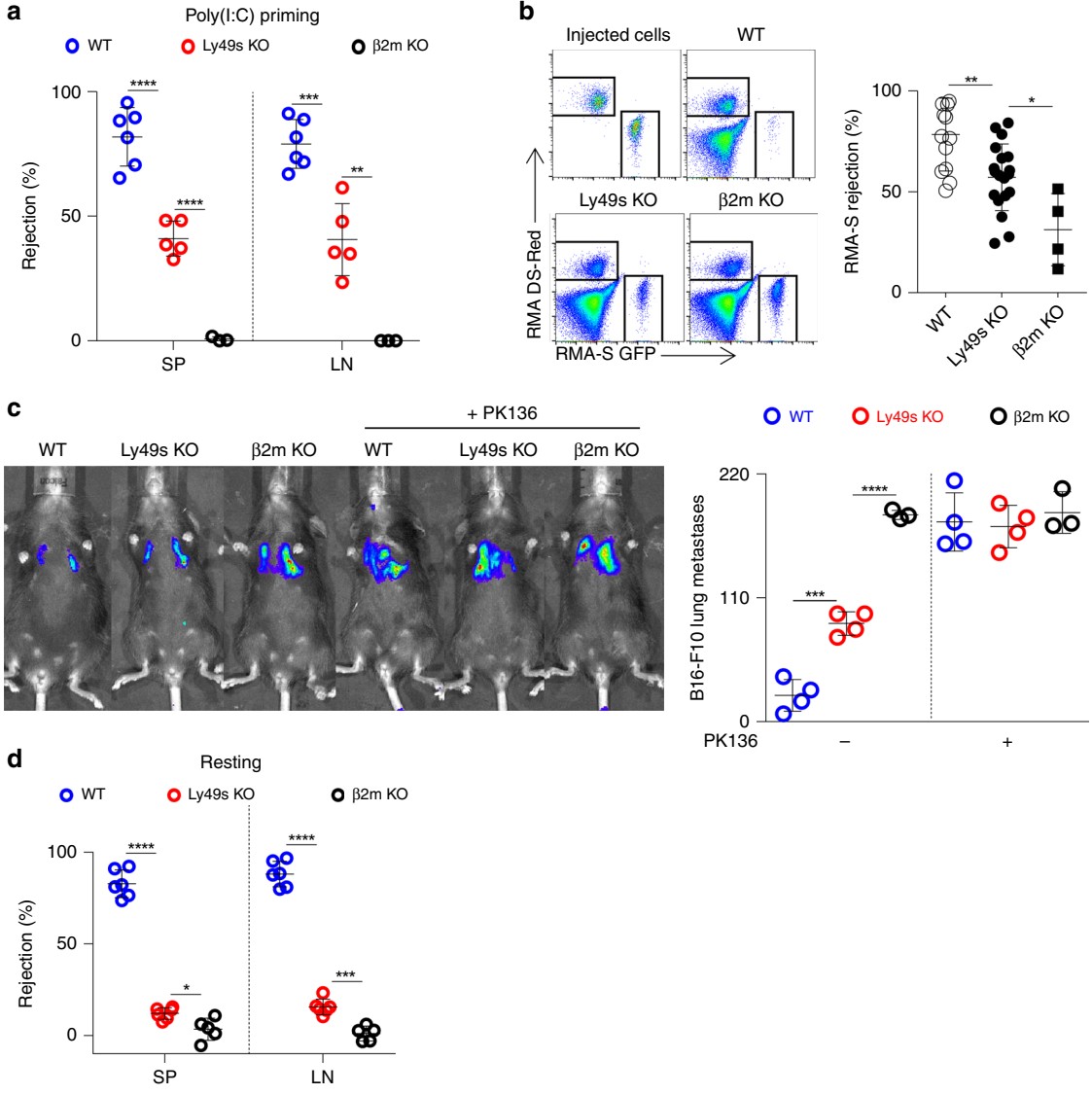

**Fig. 3** Ly49 family-deficient mice display moderate defect in NK-cell activity in vivo. **a** In vivo rejection of β2M-deficient splenocytes. High-dose CFSE-labelled β2M-deficient splenocytes were used as donor cells and low-dose CFSE-labelled WT splenocytes were used as an internal control. The percentages of rejected β2M-deficient splenocytes were calculated from the spleen (SP) and lymph nodes (LN) of the indicated mice pre-treated with poly (I:C). **b** In vivo peritoneal clearance of RMA-S cells. Representative flow cytometry plots (left panel) and the percent rejection (right panel) of RMA-S cells injected in the peritoneal cavities of the indicated mice (4–19 mice pooled from three independent experiments). **c** In vivo bioluminescence imaging of pulmonary metastases (left panel) and tumour nodules (right panel) on the lungs of the indicated mice treated with ( + ) or without (−) anti-NKR-PR1 antibody at 14 days after the B16-F10 cells inoculation. **d** Similar to (**a**), but the percentages of rejected β2M-deficient splenocytes were calculated from spleen (SP) and Lymph nodes (LN) of indicated mice without poly(I:C) treatment and were checked at 48 h post injection. Each symbol represents an individual mouse. Data shown represent two (**c**) or at least three (**a, d**) independent experiments. Mean ± SD is shown. *$p < 0.05$, **$p < 0.01$, ***$p < 0.001$ and ****$p < 0.0001$. Unpaired Student's t-tests (two-tailed) was used to calculate these values. Source data are provided as a Source Data file

under missing-self conditions. Next, we examined the ability of Ly49-deficient NK cells to control B16 melanoma metastasis. Mice were intravenously injected with B16 cells and tumour nodules on the lungs were assessed after two weeks. Compared with WT mice, Ly49s KO mice displayed an increased number of metastatic B16 colonies in the lungs. The $β2m^{-/-}$ mice displayed more substantial lung metastasis. By depleting NK cells using anti-NKR-P1C antibody, we further proved that the discrepancy among the three genotypes was largely due to the variations in NK-cell activity (Fig. 3c). Thus, we reveal a moderate defect in NK-cell function in poly(I:C)-primed Ly49s KO mice.

Because environmental stimuli such as cytokines can affect NK-cell education, we tested the in vivo activity of Ly49-deficient NK cells without poly(I:C) priming. We observed that in this

resting state, Ly49 deficiency severely compromised the ability of NK cells to reject missing-self target cells in vivo, but the difference between Ly49s KO and $β2m^{-/-}$ mice in resting state was not as pronounced as that in poly(I:C) treatment (Fig. 3d). Thus, Ly49 family is essential for NK-cell education in the resting status, but environmental stimuli likely mask the effect of this family.

**Inhibitory Ly49 members are critical for NK-cell education.** We realized that the 1.4 Mb of genomic DNA that was deleted in Ly49s KO mice comprised nine inhibitory Ly49 receptors and two activating Ly49 receptors, Ly49D and Ly49H, both of which were separated by a large amount of non-coding sequences. We

wondered whether the compromised NK-cell activity in Ly49s KO mice was due to the loss of the activating receptors Ly49D and H during the NK-cell effector phase and/or the lack of inhibitory receptors during NK-cell education. Therefore, we next dissected which key receptors accounted for the defective NK-cell activity. We initially generated Ly49D/H-deficient mice (referred to as Ly49D/H dKO mice) using CRISPR-Cas9-mediated targeting of the *Klra4* and *Klra8* genes to avoid the interference from non-coding sequences (Supplementary Fig. 3a). The absence of Ly49D and Ly49H was confirmed using flow cytometry (Supplementary Fig. 3b). The efficiency of Ly49H deletion was further verified by a functional assay in which the ectopic expression of m157, a ligand for Ly49H, could trigger the activation of WT, but not Ly49D/H-deficient NK cells (Supplementary Fig. 3c). Before the analysis of NK-cell activity in these mice, we noticed that the absence of Ly49D and Ly49H did not dramatically disturb NK-cell development (Supplementary Fig. 3d, e). Importantly, Ly49D/H dKO mice exhibited normal activation in response to the in vitro stimulation by many cell targets (Supplementary Fig. 3f), and they could effectively eliminate nearly all of the β2M-deficient splenocytes, similar to WT mice (Supplementary Fig. 3g). Hence, the loss of the activating receptors Ly49D and Ly49H does not affect NK-cell responsiveness.

Similarly, we next used the CRISPR-Cas9 technology to individually disrupt the inhibitory Ly49 receptors Ly49C and Ly49I on the C57BL/6 background (Fig. 4a). We confirmed the deletion of Ly49C/I receptors using flow cytometry (Fig. 4b and Supplementary Fig. 4a). Genomic DNA sequencing revealed a frameshift mutation in *Klra3* (Supplementary Fig. 4b). Notably, NK-cell development was largely intact in Ly49C/I dKO mice (Supplementary Fig. 4c, d). We also noticed that Ly49A+ NK cells were decreased in Ly49C/I dKO mice even though the expression level of Ly49A on individual NK cell was not influenced (Supplementary Fig. 4e). After stimulation with B16, RMA-S or YAC-1 cells, NK cells isolated from poly(I:C)-treated Ly49C/I dKO mice exhibited moderate defects in IFN-γ production and CD107a release to an extent that was comparable to Ly49s KO mice (Fig. 4c). Consistent with these findings, Ly49C/I dKO mice were not able to efficiently eliminate β2M-deficient splenocytes, either with or without poly(I:C) priming (Fig. 4d, e). These results highlight the importance of Ly49C/I in NK-cell education.

**Ly49 deletion fails to rescue β2M-null NK-cell activity**. Because the inhibitory Ly49-family receptors are required for NK-cell education by engaging with self-MHC-I molecules, we wondered why the defective phenotypes of NK cells in $β2m^{-/-}$ mice were severer than Ly49C/I dKO mice or Ly49s KO mice particularly in poly(I:C) priming state. Inhibitory Ly49 receptors, such as Ly49A, Ly49C, Ly49I and Ly49G, are expressed at high levels on NK cells from MHC-I-deficient mice[39,40]. This observation was further verified in our current study. Inhibitory Ly49 receptors were upregulated and activating Ly49 receptors tended to be downregulated (Fig. 5a and Supplementary Fig. 5a). We postulated that the loss of MHC-I not only impairs NK-cell education but also might suppress NK-cell activation during the effector phase due to the excessive expression of inhibitory Ly49 receptors. To exclude this possibility, we deleted all Ly49-family receptors in $β2m^{-/-}$ mice by intercrossing the two strains. We found that the removal of Ly49-family receptors failed to rescue the impaired responses of β2M-deficient NK cells. Mice lacking β2M and all Ly49 receptors still exhibited a severe defect following stimulation with RMA-S or YAC-1 cells that was comparable to $β2m^{-/-}$ mice (Fig. 5b). Ly49- and β2M-deficient mice also did not eliminate the splenocytes from $β2m^{-/-}$ mice in vivo (Fig. 5c). Therefore, the discrepancy between Ly49s KO mice and $β2m^{-/-}$

mice is not due to the increased expression of inhibitory Ly49 receptors in compensation for the self-MHC-I deficiency.

**NKG2A deficiency mildly impairs NK-cell activity**. The alternative possibility is that MHC-I delivers Ly49 family-independent inhibitory signals that educate NK cells. In addition to Ly49-family receptors, NKG2A expression on NK cells also strongly correlates with NK-cell responsiveness[41]. We confirmed in our study that upon YAC-1 stimulation, IFN-γ-producing NK cells among NKG2A+ population were significantly more than that in NKG2A− population (Fig. 6a). However, the lack of Ly49s or MHC-I molecules differentially affected the function of NKG2A+ and NKG2A− NK-cell populations. MHC-I deficiency pronouncedly decreased the functions of both populations in $β2m^{-/-}$ mice. Notably, although the expression of NKG2A on NK cells was not changed in Ly49s KO NK cells (Fig. 1g), the deletion of Ly49s only moderately reduced the responsiveness of NKG2A− NK cells, but it slightly but significantly increased the function of NKG2A+ NK cells (Fig. 6a). These data suggest that NKG2A likely compensate for the deficiency of Ly49s. We further evaluated the ratio of NKG2A-mediated licencing among three genotype mice and noticed that activating receptor NKR-P1C became more licenced by NKG2A in Ly49s KO mice, but not in $β2m^{-/-}$ mice (Fig. 6b). This hyper-responsiveness was not due to the altered functional machinery in Ly49s KO mice (Fig. 6c).

NKG2A binds to non-classical MHC-Ib, also known as Qa1, to inhibit NK-cell activation. We hypothesized that NKG2A is likely involved in NK-cell education, particularly when Ly49s are absent. In the end, we generated NKG2A-deficient ($Klrc1^{-/-}$) mice on a pure C57BL/6 background using the CRISPR/Cas9 technique, which was confirmed by flow cytometry (Fig. 6d). After stimulation with three types of target cells, $Klrc1^{-/-}$ NK cells exhibited very mild defects in IFN-γ secretion and degranulation response (Fig. 6e). We also found the mild defect of NK cells to response YAC-1 in a developmentally matched comparison (Supplementary Fig. 6a). In addition, NK-cell number and differentiation were nearly normal (Supplementary Fig. 6b, c). The expression of NKR-PR1 was not significantly altered (Supplementary Fig. 6d). These results excluded the possibility that the tiny functional defect in NKG2A-deficient NK cells was due to their altered development. The in vivo relevance of NKG2A deficiency was then examined and we found that poly(I:C)-primed $Klrc1^{-/-}$ mice exhibited a minimal defect to reject these self-MHC-I-deficient haematopoietic grafts (Fig. 6f). However, a slight reduction in NK-cell rejection was perceived in $Klrc1^{-/-}$ mice that was not treated with poly(I:C) (Fig. 6g), Nevertheless, this reduction was much less pronounced than that in Ly49s KO mice. Thus, NKG2A is likely needed, but not essential, for NK-cell education in resting state.

**NKG2A and Ly49s synergizes to regulate NK-cell education**. We then performed a second round of genome editing to eliminate NKG2A from Ly49s KO mice. We successfully obtained mice lacking both NKG2A and the Ly49 family (referred to as NKG2A & Ly49s dKO mice). The expression of Ly49-family receptors and NKG2A on NK cells from these mutant lines was routinely examined using flow cytometry (Supplementary Fig. 7a, b). Interestingly, this combined deficiency further impaired the expression of KLRG1 expression (Supplementary Fig. 7c) but NK-cell development was not largely disrupted in the mice lacking NKG2A and Ly49 family (Supplementary Fig. 7d, e). Notably, compared with Ly49 deletion, the simultaneous deletion of NKG2A and the Ly49 family dramatically compromised the ability of NK cells that were ex vivo primed by poly(I:C) to produce the cytokine IFN-γ and to release granules upon

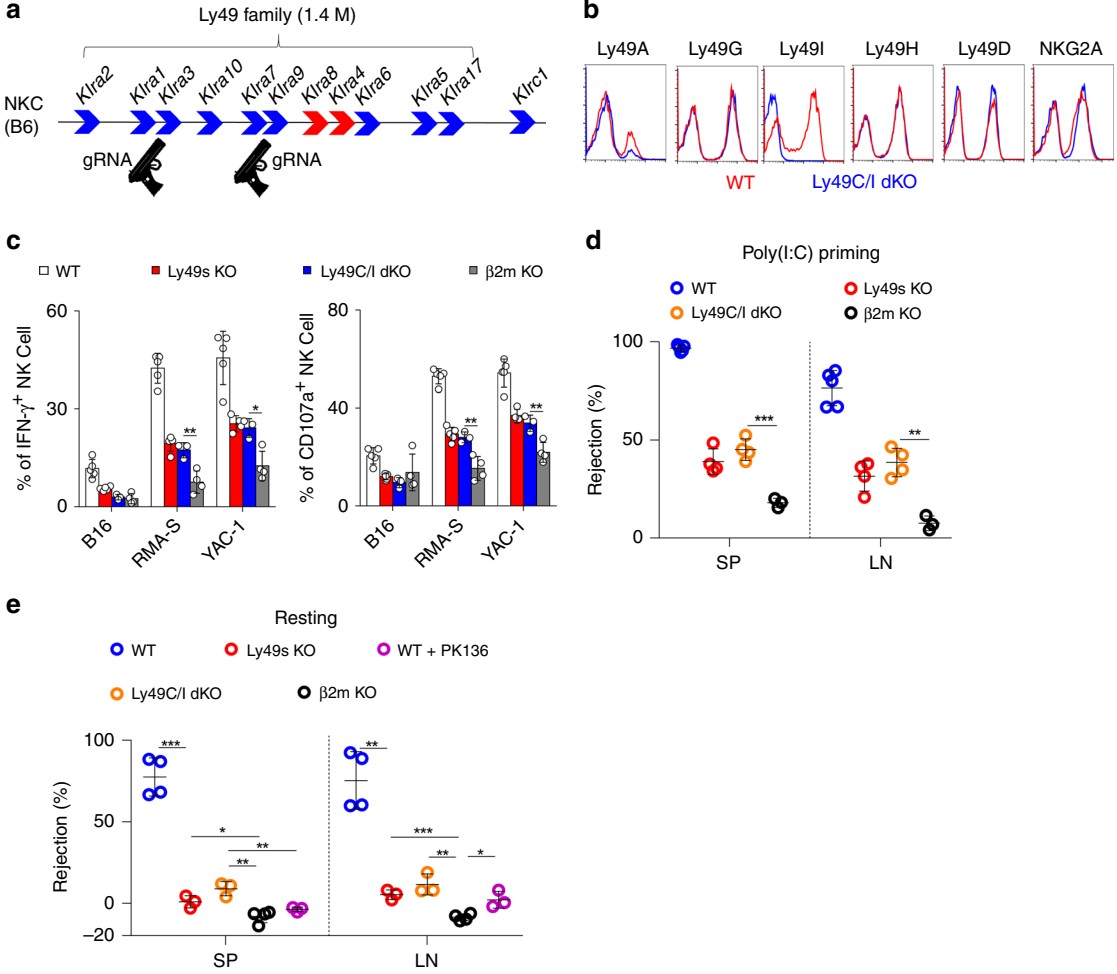

**Fig. 4** Deletion of Ly49C/I mimics the defective NK-cell activity in Ly49 family-deficient mice. **a** Diagram of the indicated genes in the NKC locus. Guns denote CRISPR gRNAs targeting *Klra3* and *Klra9*. **b** Flow cytometry analysis of the expression of the indicated receptors on splenic CD3⁻NKp46⁺ NK cells from WT (red line) and Ly49C/I dKO (blue line) mice. **c** Poly(I:C)-primed splenocytes from the indicated mice were stimulated with tumour target cells. Percentages of IFN-γ⁺ (left panel) and CD107a⁺ (right panel) gated CD3⁻NKp46⁺ NK cells were analysed. **d** In vivo rejection of β2M-deficient splenocytes (similar to Fig. 3a). **e** In vivo rejection of β2M-deficient splenocytes (similar to Fig. 3d). Each symbol represents an individual mouse. Data shown represent two (**e**) or at least three (**c**, **d**) independent experiments. Mean ± SD is shown. *$p < 0.05$, **$p < 0.01$, ***$p < 0.001$ and ****$p < 0.0001$. Unpaired Student's *t*-tests (two-tailed) was used to calculate these values. Source data are provided as a Source Data file

encountering the cellular targets, to the extent nearly equivalent to *β2m⁻/⁻* mice (Fig. 7a). This severe defect was perceived in all NK-cell subsets in a developmentally matched comparison (Fig. 7b).

To avoid the limitations associated with the deletion of the large fragment in Ly49s KO mice, we tried to generate mice that lacked nearly all of the known self-MHC-I-reactive inhibitory receptors, including Ly49C, I and NKG2A (Supplementary Fig. 8a). The intended mouse was successfully obtained, but there was additional off-target at Ly49G (Supplementary Fig. 8b). Because Ly49G does not recognize the MHC-I molecules of B6 mice[24] and plays dispensable role in NK-cell education in B6 mice (Supplementary Fig. 8c), we kept the line (referred to as NKG2A & Ly49C/G/I KO mice) for the following experiments. As expected, the ability of NK cells to respond to stimulation with cellular targets was remarkably impaired in NKG2A & Ly49C/G/I KO mice, and the defect was much severer than Ly49s KO mice, nearly equivalent to *β2m⁻/⁻* mice (Fig. 7c). Through a developmentally matched comparison, we concluded that the defective NK-cell responsiveness was not due to the change of NK-cell subsets (Fig. 7d and Supplementary Fig. 8d, e). The following in vivo assay further confirmed that the ability of NK

cells to reject "missing-self" targets was minimal in two genotypes, NKG2A & Ly49s dKO and NKG2A & Ly49C/G/I KO, either poly(I:C) priming or resting state (Fig. 7e, f). Therefore, both inhibitory Ly49 receptors and NKG2A are required for NK-cell education.

## Discussion
NK cells are presumably educated by the engagement of inhibitory receptors with self-MHC-I molecules. By making an NKC-knockdown mouse, Makrigiannis' group provided first genetic evidence supporting the critical role of Ly49 family for NK-cell education[15,30,33,42–44]. Here, we generated a serial of new genetically engineered mice that differentially lacking Ly49 family, and provided further evidence supporting the hypothesis that classic MHC-Ia molecules are required for NK-cell education by engaging the inhibitory Ly49-family receptors, at least Ly49C and/or I. Therefore, these inhibitory receptors not only suppress NK-cell function during the effector phase but they can also endow NK cells with the capability to response to the cellular targets under "missing-self" conditions. Consistent with our findings, human NK cells that express an activating receptor but

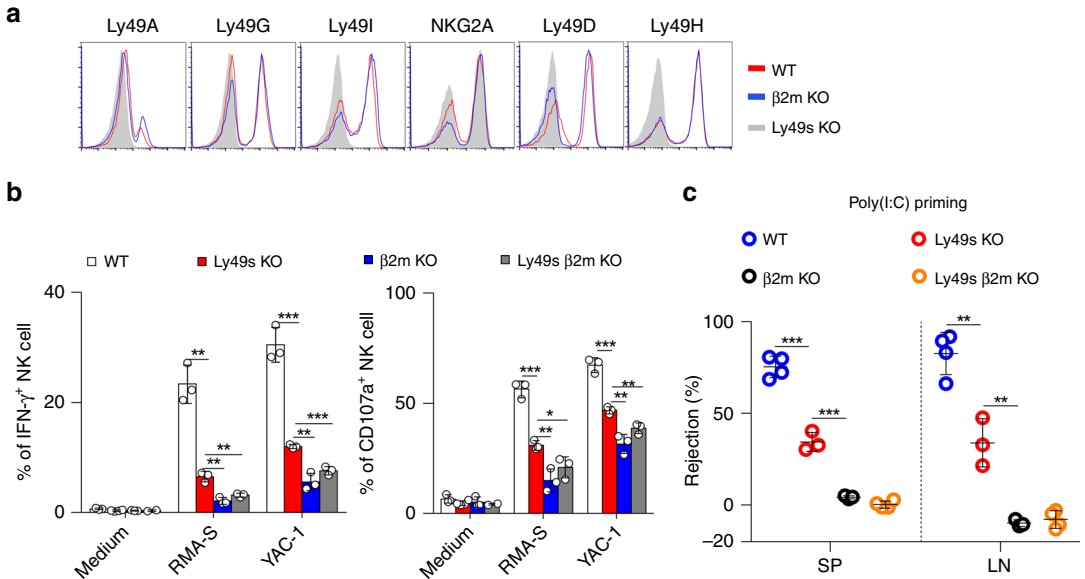

**Fig. 5** Ly49-family deletion does not rescue β2M-null NK-cell function. **a** Flow cytometry analysis of the expression of Ly49-family receptors and NKG2A on splenic CD3⁻NKp46⁺ NK cells from WT (red line) and *β2m⁻ᐟ⁻* (blue line) mice; Ly49s KO mice (grey filled curve) served as the negative control. **b** Poly(I:C)-primed splenocytes from the indicated mice were stimulated with tumour target cells. Percentages of IFN-γ⁺ (left panel) and CD107a⁺ (right panel) gated CD3⁻NKp46⁺ NK cells were analysed. **c** In vivo rejection of β2M-deficient splenocytes (similar to Fig. 3a). Each symbol represents an individual mouse. Data shown represent two (**c**) or at least three (**b**) independent experiments. Mean ± SD is shown. *p < 0.05, **p < 0.01, ***p < 0.001 and ****p < 0.0001. Unpaired Student's t-tests (two-tailed) was used to calculate these values. Source data are provided as a Source Data file

no inhibitory receptors for self-MHC class I molecules exhibit a hypo-responsive phenotype[30].

However, the inactivation of the genes encoding Ly49 inhibitory receptors did not completely prevent the development of competence in NK cells, largely due to the expression of NKG2A on NK cells. Thus, we revealed a severe NK-cell defect in mice with a combined deletion of Ly49C/I and NKG2A. Despite of this, it remains to investigate whether NKG2A is required for NK-cell education through directly interacting with non-classical MHC-Ib. Although NKG2A is usually coupled with CD94, mice lacking CD94 do not exhibited the slightly defective NK-cell responsiveness that was observed in NKG2A-deficient mice[45]. We think that NKG2A is not the sole receptor that couple with CD94. NK cells from CD94-deficient mice not only display no or low level of NKG2A, but also have significant decrease in two activating receptors, NKG2C and NKG2E. These NK-cell activating receptors are likely involved in NK-cell functional acquisition. In terms of the genetic background, because CD94 is closely linked in the NKC, 129/SvJ-originated CD94-deficient mice still preserve the NKC from 129/SvJ even several generations of backcross. From our point of view, this line is hardly comparable to the NKG2A KO mice that are pure derivation of C57BL/6.

We report here that the deletion of Ly49 family mildly affect multiple NK-cell receptors. The upregulation of NKR-P1C expression was showed in Ly49-deficient mice. On the one hand, inhibitory Ly49 signalling may influence NK-cell receptor repertoire, in particular NKR-P1C that is genetically linked in the NKC. On the other hand, NKR-P1C is probably elevated as a compensation for the loss of activating Ly49D or Ly49H. Nevertheless, because NKR-P1C is an activating receptor, we believe that the increased expression of NKR-P1C does not affect the interpretation of our findings that NK cells reject less MHC-deficient cells in vivo. In contrast, activating receptor NKG2D that is also linked in the NKC was decreased in Ly49s KO mice. We speculate that the reduced expression of NKG2D was not due to the increased transcription of *Klrk1* or the chromosome

relocation caused by large fragment deletion. It remains unclear whether the decreased NKG2D expression correlates with the subtle change of NK-cell differentiation. We also found that the percentage of Ly49A⁺ NK cells was decreased in Ly49C/I dKO mice. Because the stochastic expression of Ly49 receptors is under the control of alternate upstream promoter, named Pro1[46], we speculate that the reduced expression of these receptors may be due to the reduced transcriptional activity caused by the deficiency of Ly49C and/or Ly49I.

NK-cell responsiveness in NKG2A & Ly49C/G/I KO mice is almost comparable to *β2m⁻ᐟ⁻* mice, seemingly suggesting essential roles for these three receptors in NK-cell education. NK cells also express high levels of other inhibitory receptors within the NKC locus, which potentially bind to MHC-I. Numerous studies support the hypothesis that the functional education of NK cells mediated by self-MHC-I may be quantitative in nature[47–51]. NK cells may require a threshold of inhibitory signalling mediated by MHC-I. The residual inhibitory signalling in NKG2A & Ly49C/G/I KO mice may be not sufficient for NK-cell education. This finding partially explains why the individual deletion of Ly49-family receptors only moderately affected NK-cell function.

To date, many models have been developed to explain NK-cell education by MHC-I molecules, such as the arming model, disarming model and rheostat model[49,50,52–55]. Our current data may support the rheostat model, which argues a dynamic view of NK-cell education that is based on the strength of the inhibitory signal during NK-cell education[49,56]. The responding NK-cell balances its activation threshold as a rheostat, which allows the acquisition of NK-cell functions to be optimally adjusted by the inhibitory input. A remaining question to resolve is how inhibitory and activating signalling pathways are functionally integrated during NK-cell education and how this crosstalk may result in the dynamic thresholds that have been proposed. We previously identified signalling lymphocyte activation molecule (SLAM) family receptors as a type of endogenous self-specific activating receptor that is required for NK-cell education[38]. Further

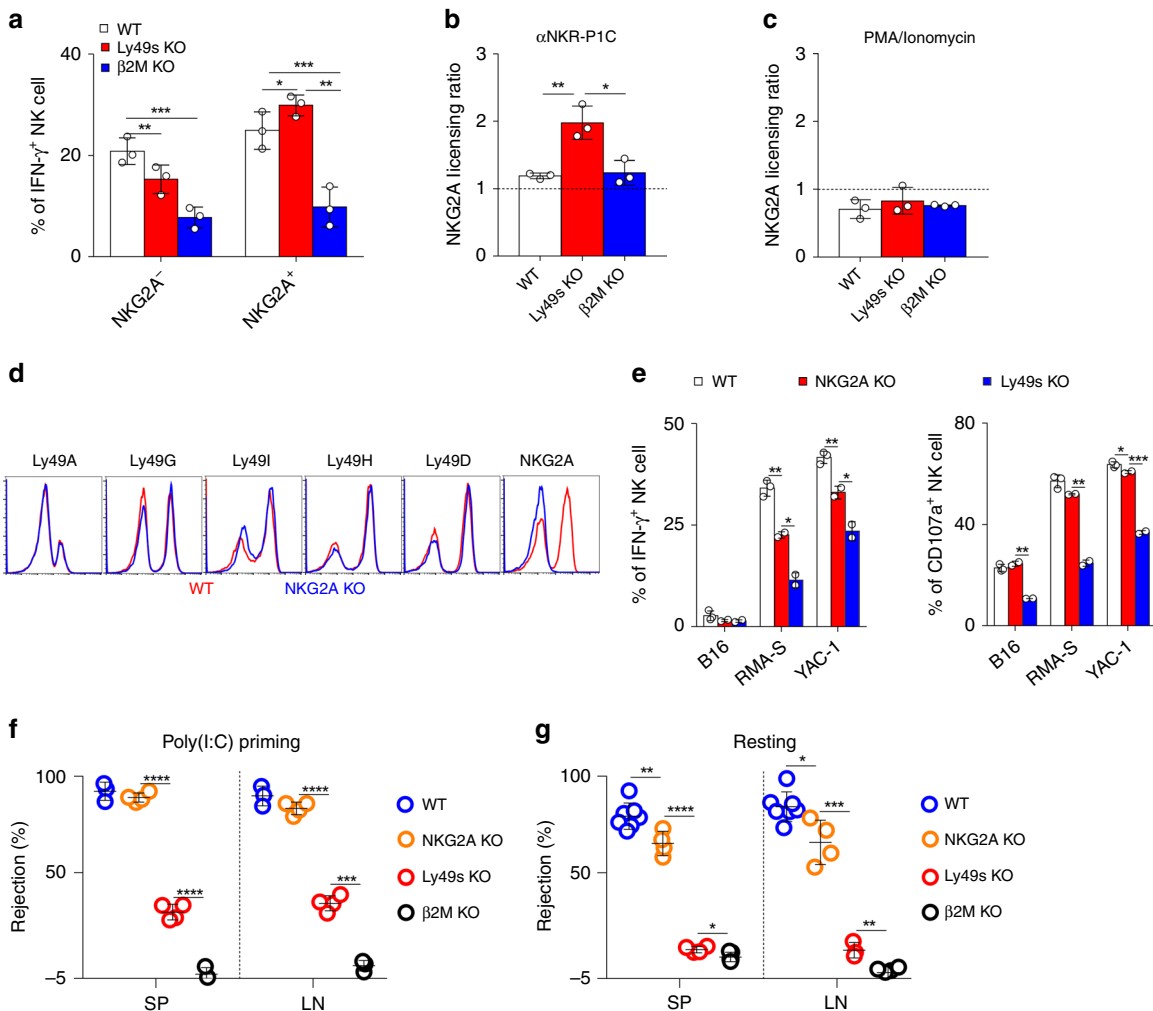

**Fig. 6** NKG2A-deficient mice display minor defects in NK-cell activity. **a** Poly(I:C)-primed splenocytes from the indicated mice were stimulated with plate-coated anti-NKR-P1C. Percentages of IFN-γ⁺ gated CD3⁻NKp46⁺ NK cells were analysed. **b, c** Licensing ratio of NKG2A for stimulation with plate-coated anti-NKR-P1C (**b**) or PMA and ionomycin (**c**). Licensing ratio is relative production of IFN-γ by NKG2A⁺ cells within gated CD3⁻NKp46⁺ NK cells comparing to NKG2A⁻ cells within gated NK cells from indicated mice. **d** Flow cytometry analysis of expression of the indicated receptors on splenic CD3⁻NKp46⁺ NK cells from WT (red line) and $Klrc1^{-/-}$ (blue line) mice (shown as NKG2A KO mice). **e** Poly(I:C)-primed splenocytes from the indicated mice were stimulated with tumour target cells. Percentages of IFN-γ⁺ gated CD3⁻NKp46⁺ NK cells were analysed. **f** In vivo rejection of β2M-deficient splenocytes (similar to Fig. 3a). **g** In vivo rejection of β2M-deficient splenocytes (similar to Fig. 3d). Each symbol represents an individual mouse. Data shown represent two (**f, g**) or at least three (**a–c, e**) independent experiments. Mean ± SD is shown. *$p < 0.05$, **$p < 0.01$, ***$p < 0.001$ and ****$p < 0.0001$. Unpaired Student's $t$-tests (two-tailed) was used to calculate these values. Source data are provided as a Source Data file

investigations are needed to determine whether these MHC-I-specific inhibitory receptors on NK cells work together with activating receptors to determine an optimal set point that depends on the balance between the signalling in the normal environment originating from self-MHC-I specific inhibitory receptors and endogenous self-specific stimulatory receptors. Thus, our current model may provide an opportunity to learn more about the molecular events that occur during inhibitory or activating signalling.

In conclusion, our data demonstrate that the inhibitory members of Ly49 family are critical for NK-cell education, while NKG2A is required for NK-cell education particularly when Ly49 family is absent.

## Methods

**Mice**. C57BL/6J (B6, H-2ᵇ) mice, CD45.1 mice and β2M-deficient mice were obtained from the Jackson Laboratory. $Rag1^{-/-}γc^{-/-}$ mice were obtained by intercrossing B6.129S7-$Rag1^{tm1Mom}$/J with NSG mice (NOD.Cg-$Prkdc^{scid}Il2rg^{tm1Wjl}$/SzJ; the Jackson Laboratory) and then backcrossing with B6 mice for at

least eight generations. Mice lacking various genes were generated by injecting synthetic guide RNAs (gRNAs) with the Cas9 enzyme into pure C57BL/6J fertilized ovules and then transferring them into surrogate female mice. Ly49-deficient mice were screened by genomic PCR. The oligonucleotide sequences of primers were summarized in Supplementary Table 1. The products were electrophoretically separated on gels and visualized using a ChampGel 5000 Gel Imaging System (Beijing Sage Creation, China). Guide RNAs for generating various mutant mice were summarized in Supplementary Table 2. All mice were housed and bred in specific pathogen-free animal facilities at Tsinghua University. All experiments using mice were approved by the Animal Ethics Committee of Tsinghua University.

**Cells**. MHC class I-deficient lymphoma cell line RMA-S (originally generated by Klas Kärre lab, Karolinska Institute, Stockholm, Sweden and kindly provided by André Veillette, Clinical Research Institute of Montreal, Canada) and YAC-1 cells (thymoma, ATCC# TIB-160) and B16-F10 cells (melanoma, ATCC #CRL-6475) were cultured in RPMI 1640 medium (STEMCELL) containing 10% fetal bovine serum (FBS). For IL-2-activated NK cells, splenic NK cells were enriched with CD49b-positive selection kit (STEMCELL) and then were cultured in RPMI 1640 medium containing 20% FBS and 1000 IU per mL human IL-2 for 5 days.

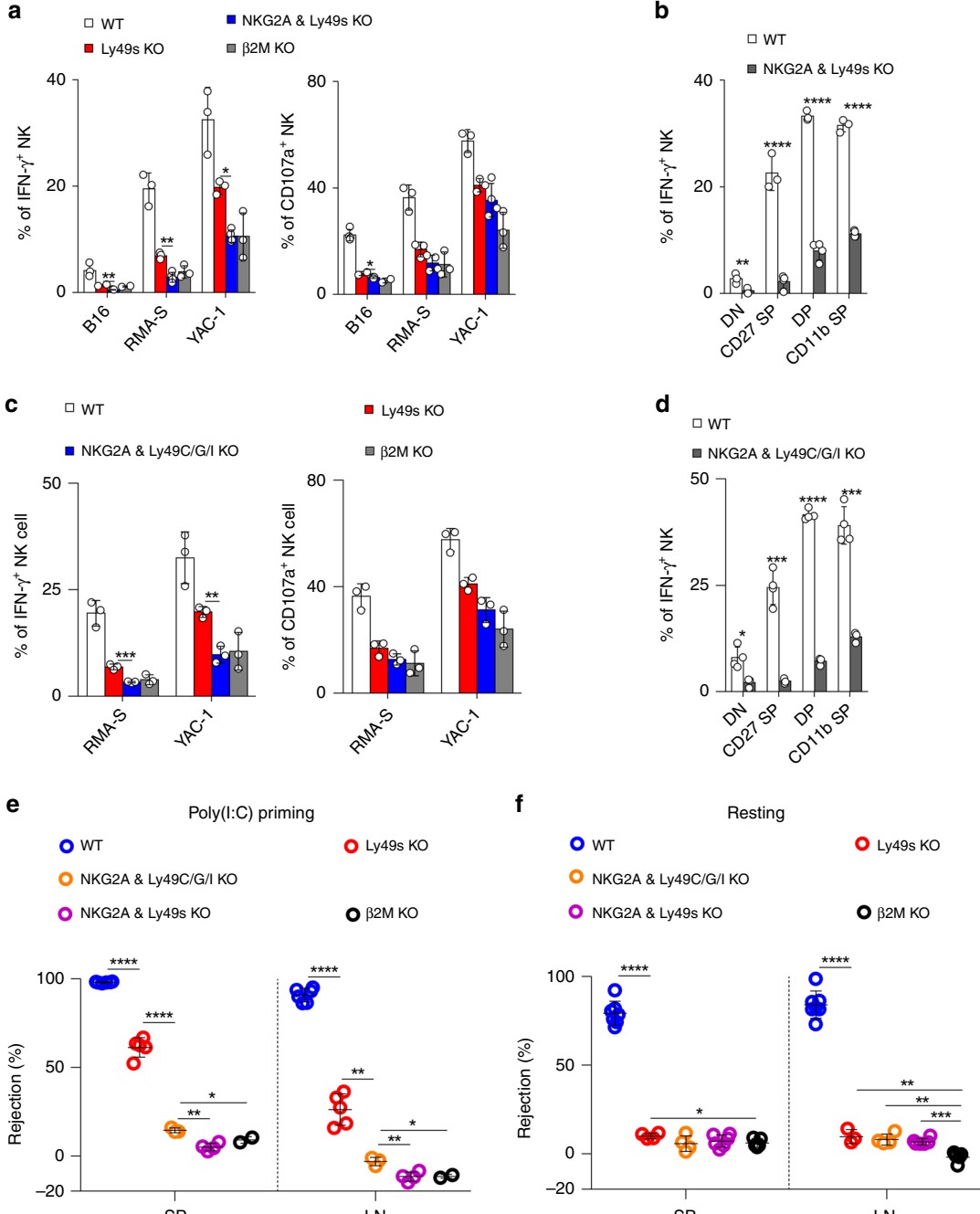

**Fig. 7** The combined deficiency of inhibitory Ly49-family receptors and NKG2A severely impairs NK-cell education. **a** Poly(I:C)-primed splenocytes from the indicated mice were stimulated with tumour target cells. The percentages of IFN-γ⁺ or CD107a⁺ gated CD3⁻NKp46⁺ NK cells were analysed. **b** Poly(I:C)-primed splenocytes from indicated mice were stimulated by YAC-1 tumour targets. Frequency of IFN-γ⁺ of four NK-cell subsets (gated CD3⁻NKp46⁺), including DN (CD27⁻CD11b⁻), CD27 SP (CD27⁺CD11b⁻), DP (CD27⁺CD11b⁺) and CD11b SP (CD27⁻CD11b⁺) were analysed. **c** Similar to (**a**). **d** Similar to (**b**). **e** In vivo rejection of β2M-deficient splenocytes (similar to Fig. 3a). **f** In vivo rejection of β2M-deficient splenocytes (similar to Fig. 3d). Each symbol represents an individual mouse. Data shown represent two (**e**, **f**) or at least three (**a–d**) independent experiments. Mean ± SD is shown. *p < 0.05, **p < 0.01, ***p < 0.001 and ****p < 0.0001. Unpaired Student's t-tests (two-tailed) was used to calculate these values. Source data are provided as a Source Data file

**Real-time quantitative PCR**. RNA was extracted from IL-2-expanded NK cells using Trizol reagent (Invitrogen) and then reverse transcribed into cDNAs using oligo d(T)ₙ primers (Promega, A3500). Real-time PCR was performed with a Bio-Rad CFX96 Real-Time System using primers specific for *Klra*-family genes. Relative quantitation of gene expression was determined after normalization to *Gapdh* expression. The following primers for Ly49-family genes were listed in Supplementary Table 3.

**Flow cytometry**. Flow cytometry was performed using a BD Fortessa flow cytometer (BD Biosciences) and data were analysed using FlowJo 10 software

(TreeStar). Monoclonal antibodies against mouse CD3 (#46-0033, clone eBio500A2, 1:200); B220 (#11-0452, clone RA3-6B2, 1:200); CD11c (#17-0114clone N418, 1:200); Gr-1 (#11-9668, 1A8-Ly6g, 1:200); F4/80 (#12-4801, clone BM8, 1:200); NKp46 (#11-3351, clone 29A1.4, 1:100); NKR-P1C (#17-5941, clone PK136, 1:200); CD49b (#12-5971, clone DX5, 1:200); CD122 (#12-1222, clone TM-b1, 1:200); CD117 (#48-1171, clone 2B8, 1:200); CD127 (#25-1273, clone SB/199, 1:200); Ly49A (#12-5856, clone A1, 1:200); Ly49I (#12-5895, clone YLI-90, 1:200); Ly49H (#17-5886, clone 3D10, 1:200); Ly49G2 (#46-5781, clone eBio4D11, 1:200); NKG2D (#25-5882, clone CX5, 1:200); NKG2A (#46-5897, clone 16a11, 1:200); DNAM-1 (#17-2261, clone 10E5, 1:200); KLRG1 (#17-5893, clone 2F1, 1:200);

CD11b (#17-0112, clone M1/70, 1:200); CD27 (#12-0271, clone LG.7F9, 1:200); MHC-I (#12-5999, clone 28-14-8, 1:200); CD45.1 (#48-0453, clone A20, 1:200); CD45.2 (#25-0454, clone 104, 1:200); IFN-γ (#12-7311, clone XMG1.2, 1:200) and CD107a (#50-1071, clone eBio1D4B, 1:100) were purchased from eBioscience. Monoclonal antibody against mouse CD49a (#74-0046, clone Ha31/8, 1:200); Ly49D (#555312, clone 4E5, 1:200) and Ly49C/I (#562055, clone 5E6, 1:200) were purchased from BD Biosciences. Flow cytometry gating strategies can be found in Supplementary Fig. 9.

**Mixed bone marrow chimera**. Donor mice were treated with 5-fluorouracil (5-FU) for 4 days. Then, bone marrow (BM) cells were separately harvested from the femurs and tibias of WT (CD45.2), Ly49s-deficient (CD45.2) and WT (CD45.1) donor mice. Next, total $5 \times 10^5$ mixed BM cells were injected into the 4.5-Gy-irradiated Rag1 γc-deficient mice through the tail vein at the ratio of 1:1 (CD45.2 to CD45.1). The spleen and bone marrow cells from the chimeric mice were analysed 8 weeks after transplantation.

**In vitro NK-cell assays**. Poly(I:C)- or IL-2-activated splenocytes or naive splenocytes ($2 \times 10^6$) were co-cultured with suspensions of different target cells ($1 \times 10^6$) or pre-seeded adherent cells in the presence of GolgiStop (BD, Biosciences) and anti-CD107a (clone eBio1D4B). For antibody stimulation, purified anti-NKR-P1C (clone PK136) was pre-coated on the 24-well plate at the indicated concentration overnight at 4 °C before adding splenocytes. After co-culturing for 6 h, cells were harvested and stained with the indicated antibodies, fixed, permeabilized with Cytofix/Cytoperm Buffer (BD Biosciences) and then stained with an anti-IFN-γ (clone XMG1.2) antibody.

**In vivo β2M-deficient splenocyte rejection assay**. Splenocytes from β2M-deficient or WT mice were labelled with 5 or 0.5 μM CFSE (carboxyfluorescein diacetate succinimidyl ester; Invitrogen), respectively. Equivalent numbers ($1 \times 10^6$) of each cell population were mixed and intravenously injected. The recipient mice were intraperitonally injected with the poly(I:C) for 18 h before transferring labelled cells. After transferring for 6 h, CFSE$^+$ cells were identified from the spleen and lymph node using flow cytometry. For the recipient mice without poly(I:C) treatment[57–59], CFSE$^+$ cells were identified from the spleen and lymph node at 48 h after transferring. The percentage of β2M-deficient splenocytes rejection was calculated as following: $100 \times [1 - ($percentage of residual β2M-deficient splenocytes in total CFSE$^+$ cells of experimental group/percentage of β2M-deficient splenocytes in total CFSE$^+$ cells of injected cells)].

**In vivo RMA-S cell clearance assay**. Mice that had been treated with 200 μg of poly(I:C) for 18 h were intraperitoneally injected with a mixture of GFP-expressing RMA-S cells ($1 \times 10^6$) and DsRed-expressing RMA cells ($1 \times 10^6$). After 18 h, mice were euthanized and peritoneal cells were collected with PBS. The percentage of RMA-S cell rejection was calculated using the following formula: $100 \times [1 - ($percentage of the residual GFP$^+$ population among the total GFP$^+$ and DsRed$^+$ cells in the experimental group/percentage of the residual GFP$^+$ population among the total GFP$^+$ and DsRed$^+$ cells in the Rag1 γc-deficient mice)].

**B16 pulmonary metastasis model**. Mice were intravenously injected with $2 \times 10^5$ B16-F10 cells that express luciferase through the tail vein. In vivo bioluminescence imaging of pulmonary metastases were performed 14 days after tumour challenge. Avertin (Sigma) was intraperitoneally injected into mice at a dose of 375 mg per kg body weight to induce anaesthesia. Mice were administered intraperitoneal injections of 20 mg per mL D-luciferin at a dose of 150 mg per kg body weight for 8–10 min; then, imaging was performed with a Lumina II instrument (PerkinElmer). After bioluminescence imaging, mice were sacrificed and the number of tumour nodules on the lungs was counted and the lungs were weighed.

**Statistics**. Prism 7 software was used to perform unpaired Student's $t$ tests (two-tailed). A $p$ value of <0.05 was considered significant, $*p < 0.05$, $**p < 0.01$, $***p < 0.001$ and $****p < 0.0001$. Data were presented as the mean ± SD.

**Reporting summary**. Further information on research design is available in the Nature Research Reporting Summary linked to this article.

## Data availability
No datasets were generated or analysed during the current study. The source data underlying Figs. 1d, f, g, i, j, 2a–f, 3a–d, 4c–e, 5b, 6a–c, e–g, 7a–f and supplementary Figs. 1c–e, 2a, 3c–g, 4c–e, 5a, 6a–d, 7c–e and 8c–e are provided as a Source Data file.

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

## Acknowledgements

We acknowledge Dr. Zai Chang and Jing Zhang (Animal Facility of Tsinghua University, China) for their help of preparation of gene-modified mice. This work is supported by National Key Research & Developmental Program of China (2018YFC1003900) and Natural Science Foundation of China (81725007, 31830027 and 31821003) and Beijing Natural Science Foundation (5172018).

## Author contributions

X.Z. and J.F. conceived the project, designed and performed most experiments. S.C. and H.Y. conceived the study. Z.D. initiated and supervised the project and wrote the paper with the help of X.Z and J.F.

## Competing interests

The authors declare no competing interests.

## Additional information

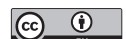

