## [Peer Review File · Nature Communications]

Reviewers' comments:

Reviewer #1 (NK function, NK receptors)(Remarks to the Author):

To study the role of Ly49 receptors in NK cell licensing, Zhang et al. generated several new mouse lines using CRISPR/Cas technology in fertilized oocytes. The data show that the functional activities of NK cells from mice with an entire deletion of all Ly49 genes were only impaired moderately when tested in ex vivo and in vivo assays. The same conclusion was reached when mice only deficient in Ly49C and Ly49I were analyzed. NK cells from mice deficient in the two activating receptors Ly49D and Ly49H were not impaired in the assays examined here. Most interestingly, NK cells from mice deficient in Ly49C/I and NKG2A were impaired as strongly as NK cells from beta2-deficient mice indicating additive effects of these two types of inhibitory MHC-specific receptors in NK cell licensing.

The authors made a substantial effort to define the requirements for NK cell licensing by generating several new mouse lines. The experiments assessing NK cell function are straight forward and well controlled. The data are novel since mice with the reported genotypes have not yet been described. In particular, the finding that loss of Ly49C/I led to the same effect as the complete deletion of all L49 genes is quite remarkable. Similarly, the observation that NKG2a deficiency led to an additive effect in this setting is intriguing. In sum, the paper is of significant interest for many people working in the NK cell field. There are a few points the authors may want to address to improve the quality of their study.

1) The authors "use" 4 figures (Fig.1-Fig.4) to present the data from the Ly49s KO mice. However, the findings from the other 5 mouse strains including the more interesting ones are condensed in the remaining 2 figures. I suggest that the figures are rearranged to achieve a more relevant distribution. For this, some of data from the Ly49s KO mice (e.g. Fig.1B, Fig.1C, Fig.4C, Fig.3C) could also be shown in supporting information.

2) Percentages and absolute numbers of NK cells in Ly49s KO mice are increased compared to controls (Fig. 2A/B). In the bone marrow chimeric mice, however, this difference is no longer evident. The authors should state whether this discrepancy was observed in more than one independent experiment. More generally spoken, in each figure, the number of independent experiment should be indicated.

3) The authors show NK cell numbers and extensive phenotypic analysis from Ly49s KO mice but the corresponding data from the more interesting NKG2A&Ly49s KO and NKG2A&Ly49C/G/I mice that exhibit a more severe defect in NK cell function are missing. Why? This point is also important to exclude the possibility that reduced NK cell numbers in these mice were responsible for lack of cell rejection in the functional in vivo assay.

Reviewer #2 (NK biology, NK therapy, ILC)(Remarks to the Author):

This is an interesting article from the Dong laboratory that reports the generation of CRISPR/Cas9-engineered mice that lack all members of the Ly49 gene family. Using these unique models, the authors could examine the phenotype as well as the in vitro and in vivo reactivity of NK cells in order to test the role of inhibitory MHC class I in NK cell education. It is obvious that most of the interest of the present paper relies on the comparison of NK cell reactivity between WT mice, MHC class I-deficient mice and mice that are deficient for the types of inhibitory MHC class I receptors, Ly49 and

NKG2A. This is a technical tour-de-force and a timely study. However, some parts of the present report are raising major concerns as listed below.

1. A decrease in the size of the KLRG1+ NK cell subset regardless of CD11bCD27 maturation phenotype has been consistently observed in MHC class I-deficient mice. It is surprising that this phenotype which is detected in Ly49-deficient is not highlighted, in the text as it should be (despite the statistical difference in Fig. 2C). What about the comparison with MHC class I-deficient mice and in Ly49/NKG2A-deficient mice?

2. The upregulation of NK1.1 expression in Ly49-deficient mice might be an issue for the interpretation of the in vivo experiments. Could authors comment of this and does this upregulation occur in NKG2A-deficient mice?

3. A major readout of NK cell reactivity relies on the in vivo rejection of splenocytes from MHC class I-deficient origin. The classical assay is close to what has been performed in this paper except that the measurement of injected cell ratios is performed at 20 to 48 hours post-injection and not 6 hours as reported here. The authors should analyze the presence of injected cells into recipient mice over a kinetics that is extended to these later time-points to find out whether the reactivity of NK cells in the absence of Ly49, NKG2A, Ly49 and NKG2A, or MHC Class I confirm or not the claims of the present report. In addition, classical assays do not require the priming of NK cells in the recipient mice using poly-IC; again, the standard assay should be used by the authors.

4. NK cell reactivity is dependent upon environmental cues, as illustrated by the restoration of NK cell hyporesponsiveness by cytokine treatments. To ensure the robustness of the data, it is thus critical that the comparison between the reactivity of NK cells originating from the various genotypes (absence of Ly49, NKG2A, Ly49 and NKG2A, or MHC Class I) is performed using co-housed mice or that the gut microbiota of all mice used in the experiments is analyzed and compared.

5. As beta2 microglobulin is associated with the stable cell surface expression of several molecules besides classical MHC class I and Qa1b, the authors should use KbDd double deficient mice as a control of NK cell hyporesponsiveness induced by the lack of MHC Class I recognition. This control is more adequate than beta2 microglobulin-deficient mice for the purpose of the study which is restricted to NK cell education by MHC class I molecules.

6. The statistical analysis of the results should be revisited (enough mice per group and enough repeats of the experiments?), and SD (not SEM) should be presented

Other comments

- CD94-deficient mice (hence NKG2A-deficient mice) do not show deficiencies in NK cell reactivity.

Authors should quote the papers and comment.

- In the introduction, the sentence as which Ly49 and NKG2A are redundant is both vague and inaccurate. Authors should develop their interpretation of the literature, change the phrasing or delete this sentence.

Reviewer #3 (NK development/education, NK receptors)(Remarks to the Author):

Comments to authors:

In this manuscript, Zhang and co-authors examine the role of the Ly49 receptor family in the regulation of Natural Killer (NK) cell function and education by performing CRISPR-mediated mutagenesis of genes in the Ly49 locus and NKG2A. The importance of Ly49 receptors in NK cell activity and education is known. It has been previously established that NK inhibitory receptors, such as Ly49 C, I, G, and others are important mediators of NK activity and NK education through their interaction of self-MHC-I ligands. The NK-mediated rejection of MHC-I-deficient cells is central component of the "missing self" model in NK immune function. This rejection is mediated by several NK inhibitory receptors, including Ly49 receptors.

The work presented here provides a genetic basis for Ly49 function in education and NK development. This work follows years of circumstantial evidence and agrees with findings in numerous studies in mice and human studies. Also, the importance of NKG2A was previously identified in several studies which demonstrated that CD94/NKG2A played a major role in normal NK cell education, both independent of and in coordination with expression of KIRs in humans. The authors present this current study as the "first genetic evidence supporting the hypothesis that classic MHC-Ia molecules are required for NK cell education by engaging the inhibitory Ly49 family receptors". However, previous genetic evidence by Bélanger et al., 2012 *Blood*, showed that Ly49 expression was necessary for NK-cell education to self-MHC-I through use of a genetic depletion of the Ly49 locus and a rescue of this knockdown with a Ly49I transgene. While the collection of mutants generated by Zhang et al, represent a valuable resource for the research community and an impressive capacity at genetically manipulating the Ly49 locus, the manuscript suffers heavily from a lack of many necessary controls and important experiments which are detailed below. Additionally, the authors need to sufficiently recognize the relevant work from previous studies here that agree with their own findings.

Detailed Comments:

1. Figure 1B. The PCR demonstration of the pan-Ly49 deletion is uninformative as a method and exhibits a rather unimpressive demonstration of genetic absence. The authors need to demonstrate disruption of the Ly49 directly, through use of Southern Blot (Ly49C/A would likely be sufficient to detect most Ly49s).
2. The authors assess in detail features of NK cell development in the pan-Ly49 knockout mice. However, no assessment of NK cell development is performed on other knockout mice in the study. While it is presumed the targeted knockouts of Ly49H/D, Ly49C/I, Ly49C/G/I would not produce any phenotype more severe than the pan-Ly49 knockout, it is not clear what developmental defects exist with NKG2A knockout mice. Therefore, the authors need to assess whether development of ILCs, T cells, plasmacytoid dendritic cells (pDC), neutrophils and macrophages is affected by the pan-Ly49 mutation. In Figure 1C, the expression of ly49b and ly49q is not NK specific. Ly49b is expressed in macrophages and Ly49Q is expressed on plasmacytoid dendritic cells (which likely explains why no transcript is detected for Ly49q. Authors do not examine or even discuss the implications of non-NK expression of Ly49s. Quantitative PCR of Ly49b, should be from macrophage mRNA and Ly49q from pDC mRNA (or total splenic mRNA) and should be included as a separate panel. Furthermore, Figure 1C requires proper positive and negative controls that works for both wildtype and Ly49-KO.
3. Figure 1D. Flow cytometry that measures Ly49Q on pDCs is needed.
4. Use of name NK1.1 throughout manuscript is no longer acceptable. NKRP1C is the current acceptable name.
5. Figure 2B requires proper littermate controls and should include quantification of liver NK cell population as well.
6. Authors need to address whether surface expression changes in NKG2D represent a functional consequence of Ly49 mutation or a genetic consequence caused by the significant chromosomal disruption that neighbors the NKG2D gene.
7. Authors need to examine why if NK development is largely intact as shown in Figure 2D, why is NKG2D affected?

8. Figure 2F should include measurements of T cell and B cell repopulation as a control to show that they are not affected.
9. Figure 3. NK cytotoxicity assay is needed as CD107a is not a sufficient surrogate for killing. NK cytotoxicity assay can be an in vitro assay and does not need to be radiometric. If authors insist that in vivo demonstration of killing is sufficient then they need to perform NK depletion in these mice to show a loss of killing.
10. Figure 5. Authors need to verify that Ly49H is functionally deleted. Since Ly49H interacts with MCMV protein m157, authors can demonstrate this loss of function by assaying the NK response to MCMV infection.
11. Figure 5F. One would anticipate Ly49A levels to increase under these conditions. Authors should comment on this observation.
12. Figure 5F. Authors need to show Ly49C loss by either use of anti-Ly49C/I 5E6 antibody and measuring ly49C mRNA levels.
13. Figure 5I/J. These findings are not novel and therefore if used, it should be presented as supplementary data and not presented with the main body of results.
14. Figure 5K. Missing statistical measure for comparisons between other groups.
15. Figure 6B. In the ly49C/I single knockout and NKG2A single knockout represent a great opportunity to test Ly49C/I and NKG2A roles in education. NKG2A educated mice should be have higher Ly49C/I (most NK cells are Ly49C/I positive) and vice versa. The authors need to perform an in vivo rejection assay for the NKG2A single knockout mice.
16. Figure 6I. The authors need to explain why they targeted Ly49G.
17. Figure 6F. The authors need to assess development (CD11b/27) for the total knockout generated (ly49C/G/I and NKG2A)

Point-to-point Responses to Reviewers

Reviewer #1 (NK function, NK receptors) (Remarks to the Author):

To study the role of Ly49 receptors in NK cell licensing, Zhang et al. generated several new mouse lines using CRISPR/Cas technology in fertilized oocytes. The data show that the functional activities of NK cells from mice with an entire deletion of all Ly49 genes were only impaired moderately when tested in ex vivo and in vivo assays. The same conclusion was reached when mice only deficient in Ly49C and Ly49I were analyzed. NK cells from mice deficient in the two activating receptors Ly49D and Ly49H were not impaired in the assays examined here. Most interestingly, NK cells from mice deficient in Ly49C/I and NKG2A were impaired as strongly as NK cells from beta2-deficient mice indicating additive effects of these two types of inhibitory MHC-specific receptors in NK cell licensing.

The authors made a substantial effort to define the requirements for NK cell licensing by generating several new mouse lines. The experiments assessing NK cell function are straight forward and well controlled. The data are novel since mice with the reported genotypes have not yet been described. In particular, the finding that loss of Ly49C/I led to the same effect as the complete deletion of all L49 genes is quite remarkable. Similarly, the observation that NKG2a deficiency led to an additive effect in this setting is intriguing. In sum, the paper is of significant interest for many people working in the NK cell field. There are a few points the authors may want to address to improve the quality of their study.

1) The authors “use” 4 figures (Fig.1-Fig.4) to present the data from the Ly49s KO mice. However, the findings from the other 5 mouse strains including the more interesting ones are condensed in the remaining 2 figures. I suggest that the figures are rearranged to achieve a more relevant distribution. For this, some of data from the Ly49s KO mice (e.g. Fig.1B, Fig.1C, Fig.4C, Fig.3C) could also be shown in supporting information.

A: According to the reviewer’s suggestion and our new data, the figures are rearranged. Please see related change:

1. Fig. 1B,C is moved to Supplementary Fig. 1a,c, and Fig. 3C to Supplementary Fig. 2a.
2. Fig. 1 and 2 merge into new Fig.1.
3. Fig. 5J is changed to Supplementary Fig. 5a, according to the suggestion of reviewer #3.
4. Fig. 5, 6 are rearranged into new Fig. 4-7 and supplementary Fig. 3 to achieve a more relevant distribution.

2) Percentages and absolute numbers of NK cells in Ly49s KO mice are increased compared to controls (Fig. 2A/B). In the bone marrow chimeric mice, however, this difference is no longer evident. The authors should state whether this discrepancy was observed in more than one independent experiment.

A: We appreciate the reviewer's reminder. This BM chimeric assay was done only once in the previous version. To confirm this phenotype, we repeated the experiment twice and found that Ly49-family deletion caused little difference in NK-cell development, likely suggesting an extrinsic effect. Please see the data in **Fig. 1i**

3) The authors show NK cell numbers and extensive phenotypic analysis from Ly49s KO mice but the corresponding data from the more interesting NKG2A&Ly49s KO and NKG2A&Ly49C/G/I mice that exhibit a more severe defect in NK cell function are missing. Why? This point is also important to exclude the possibility that reduced NK cell numbers in these mice were responsible for lack of cell rejection in the functional in vivo assay.

A: We provided new data showing that NK-cell number and differentiation were not dramatically altered. (Please see **Supplemental Fig. 7d, e, 8d, e**). In addition, we examined NK-cell activity in a developmentally matched comparison to avoid the potential developmental problem in several Ly49s KO mice. We got the similar conclusion that the defective NK cell function was not due to the potential alteration of NK-cell differentiation (Please see **Fig. 7b, d and Supplemental Fig. 6a**).

Reviewer #2 (NK biology, NK therapy, ILC) (Remarks to the Author):

This is an interesting article from the Dong laboratory that reports the generation of CRISPR/Cas9-engineered mice that lack all members of the Ly49 gene family. Using these unique models, the authors could examine the phenotype as well as the in vitro and in vivo reactivity of NK cells in order to test the role of inhibitory MHC class I in NK cell education. It is obvious that most of the interest of the present paper relies on the comparison of NK cell reactivity between WT mice, MHC class I-deficient mice and mice that are deficient for the types of inhibitory MHC class I receptors, Ly49 and NKG2A. This is a technical tour-de-force and a timely study. However, some parts of the present report are raising major concerns as listed below.

1. A decrease in the size of the KLRG1+ NK cell subset regardless of CD11bCD27 maturation phenotype has been consistently observed in MHC class I-deficient mice. It is surprising that this phenotype which is detected in Ly49-deficient is not highlighted, in the text as it should be (despite the statistical difference in Fig. 2C). What about the comparison with MHC class I-deficient mice and in Ly49/NKG2A-deficient mice?

A: We did not realize that the expression of KLRG1 should be mentioned in the text until we read the paper by David Raulet group¹. After re-examining the KLRG1 expression on NK cells from those requested mice, we found that KLRG1 was moderately down-regulated in Ly49s KO mice and even observed a dramatic decrease of KLRG1 expression on NK cells from NKG2A-Ly49s KO mice to an extent similar to $\beta 2m^{-/-}$ mice (Please see **Supplementary Fig. 7c**). These data suggest that the percentage of KLRG1⁺ NK cells is regulated by Ly49- and NKG2A inhibitory receptors. We change the text in the revised version.

2. The upregulation of NK1.1 expression in Ly49-deficient mice might be an issue for the interpretation of the in vivo experiments. Could authors comment of this and does this upregulation occur in NKG2A-deficient mice?

A: We did not notice a significant change of NK1.1 expression in NKG2A KO mice (**Supplementary Fig. 6d**).

We try to comment on this issue in the discussion part of the revised manuscript. On the one hand, we think that inhibitory Ly49 signaling may influence NK cell receptor repertoire, in particularly NK1.1 that

is genetically linked in the NKC. On the other hand, the up-regulation of NK1.1 may be due to the loss of activating Ly49D or H as a compensation. Nevertheless, because NK1.1 is an activating receptor, we believe that the increased expression of NK1.1 does not affect the interpretation of our findings that NK cells reject less MHC-deficient cells *in vivo*.

3. A major readout of NK cell reactivity relies on the *in vivo* rejection of splenocytes from MHC class I-deficient origin. The classical assay is close to what has been performed in this paper except that the measurement of injected cell ratios is performed at 20 to 48 hours post-injection and not 6 hours as reported here. The authors should analyze the presence of injected cells into recipient mice over a kinetics that is extended to these later time-points to find out whether the reactivity of NK cells in the absence of Ly49, NKG2A, Ly49 and NKG2A, or MHC Class I confirm or not the claims of the present report. In addition, classical assays do not require the priming of NK cells in the recipient mice using poly-IC; again, the standard assay should be used by the authors.

A: In order to select an appropriate time-point to monitor the ability of NK cells to reject targets without poly(I:C) priming, we reviewed the related literature² and then performed the *in vivo* assay in wild-type mice in a kinetic way. After it was determined that the measurement of injected cell ratios was performed at 48 hours post-injection, we confirmed that the reactivity of NK cells in the absence of Ly49, NKG2A, Ly49 and NKG2A, or β 2m was mostly consistent with our previous data (**Fig. 3d, 4e, 6g, 7f**).

4. NK cell reactivity is dependent upon environmental cues, as illustrated by the restoration of NK cell hyporesponsiveness by cytokine treatments. To ensure the robustness of the data, it is thus critical that the comparison between the reactivity of NK cells originating from the various genotypes (absence of Ly49, NKG2A, Ly49 and NKG2A, or MHC Class I) is performed using co-housed mice or that the gut microbiota of all mice used in the experiments is analyzed and compared.

A: We understand the issue. However, when various genotypes were used in an assay, we routinely co-house age-matched mice after weaning and kept in specific pathogen-free animal facilities to avoid environmental variation of mouse living.

5. As beta2 microglobulin is associated with the stable cell surface expression of several molecules besides classical MHC class I and Qa1b, the authors should use KbDd double deficient mice as a control of NK cell hyporesponsiveness induced by the lack of MHC Class I recognition. This control is more adequate than beta2 microglobulin-deficient mice for the purpose of the study which is restricted to NK cell education by MHC class I molecules.

A: We appreciate this important suggestion and totally agree with the reviewer that if K^bD^b-deficient mice were used as a control, we will go straight to test the role of the pair of inhibitory Ly49s and classical MHC-I molecule in NK cell education. However, K^bD^b-deficient mice was initially generated from 129-background embryonic stem cells^{3,4}. Therefore, we are not reluctant to compare the phenotype of this mouse. As an alternative, we are preparing new K^bD^b-deficient mice onto C57BL/6 background. Unfortunately, it will take one more year to get the mice, so we are going to compare the phenotype of this mouse in the future.

During the preparation of the revision, we found that in the resting state, Ly49 deficiency severely compromised the ability of NK cells to reject missing-self target cells in vivo, while NKG2A deletion only mildly affected NK cell function (**Fig. 3d, 5g**). Besides, under a condition without poly I:C priming, the difference was relatively less pronounced between Ly49s KO and $\beta 2m^{-/-}$ mice (**Fig. 3d**). These new data suggest that Ly49 family is essential for NK cell education in the resting status, while NKG2A play a minor role. In view of this situation, we play down the previous claim that NKG2A is required for NK cell education via binding with non-classical MHC-Ib in the revised manuscript. We expect reviewers to accept the current solution, although it is not perfect.

6. The statistical analysis of the results should be revisited (enough mice per group and enough repeats of the experiments?), and SD (not SEM) should be presented.

A: We re-examined the statistical analysis thoroughly. As advised, we changed SEM for SD.

CD94-deficient mice (hence NKG2A-deficient mice) do not show deficiencies in NK cell reactivity. Authors should quote the papers and comment.

A: As suggested, we cited the literature⁵. We discussed the difference between two genotypes in the part of Discussion. In brief, NKG2A is not the sole receptor that couple with CD94. NK cells from CD94-deficient mice not only display no or low level of NKG2A, but also have significant decrease in two activating receptors, NKG2C and NKG2E. These NK-cell activating receptors are likely involved in NK cell functional acquisition. In terms of the genetic background, because CD94 is closely linked in the NKC, 129/SvJ-originated CD94-deficient mice still preserve the NKC from 129/SvJ even several generation of backcross. From our point of view, this line is hardly comparable with the *NKG2A*^{-/-} mice that are pure derivation of C57BL/6.

In the introduction, the sentence as which Ly49 and NKG2A are redundant is both vague and inaccurate. Authors should develop their interpretation of the literature, change the phrasing or delete this sentence.

A: The sentence was deleted.

Reviewer #3 (NK development/education, NK receptors) (Remarks to the Author):

In this manuscript, Zhang and co-authors examine the role of the Ly49 receptor family in the regulation of Natural Killer (NK) cell function and education by performing CRISPR mediated mutagenesis of genes in the Ly49 locus and NKG2A. The importance of Ly49 receptors in NK cell activity and education is known. It has been previously established that NK inhibitory receptors, such as Ly49 C, I, G, and others are important mediators of NK activity and NK education through their interaction of self-MHC-I ligands. The NK mediated rejection of MHC-I-deficient cells is central component of the “missing self” model in NK immune function. This rejection is mediated by several NK inhibitory receptors, including Ly49 receptors.

The work presented here provides a genetic basis for Ly49 function in education and NK development. This work follows years of circumstantial evidence and agrees with findings in numerous studies in mice and human studies. Also, the importance of NKG2A was previously identified in several studies which

demonstrated that CD94/NKG2A played a major role in normal NK cell education, both independent of and in coordination with expression of KIRs in humans. The authors present this current study as the “first genetic evidence supporting the hypothesis that classic MHC-Ia molecules are required for NK cell education by engaging the inhibitory Ly49 family receptors”. However, previous genetic evidence by Bélanger et al., 2012 Blood, showed that Ly49 expression was necessary for NK-cell education to self-MHC-I through use of a genetic depletion of the Ly49 locus and a rescue of this knockdown with a Ly49I transgene. While the collection of mutants generated by Zhang et al, represent a valuable resource for the research community and an impressive capacity at genetically manipulating the Ly49 locus, the manuscript suffers heavily from a lack of many necessary controls and important experiments which are detailed below. Additionally, the authors need to sufficiently recognize the relevant work from previous studies here that agree with their own findings.

A: To recognize the relevant work from previous studies, we changed our text and cited the related literatures in our revised manuscript (please see Introduction and Discussion parts).

1. Figure 1B. The PCR demonstration of the pan-Ly49 deletion is uninformative as a method and exhibits a rather unimpressive demonstration of genetic absence. The authors need to demonstrate disruption of the Ly49 directly, through use of Southern Blot (Ly49C/A would likely be sufficient to detect most Ly49s).

A: To directly prove the deletion of the Ly49 family, we cloned the genomic DNA sequence between two sites that the guide RNAs target. We demonstrated that 1415735 base pairs starting from *Klra2* to *Klra17* was deleted (**Supplementary Fig. 1b**).

2. The authors assess in detail features of NK cell development in the pan-Ly49 knockout mice. However, no assessment of NK cell development is performed on other knockout mice in the study. While it is presumed the targeted knockouts of Ly49H/D, Ly49C/I, Ly49C/G/I would not produce any phenotype more severe than the pan-Ly49 knockout, it is not clear what developmental defects exist with NKG2A knockout mice. Therefore, the authors need to assess whether development of ILCs, T cells, plasmacytoid dendritic cells (pDC), neutrophils and macrophages is affected by the pan-Ly49 mutation.

A: As suggested, the requested experiments were supplied in **Supplementary Fig. 3d, e; 4c, d; 6b, c; 7d, e; 8d, e**. Briefly, NK cell development was not dramatically changed in these knockout mice. Besides, the quantification of T cells, pDC, cDC, neutrophils, macrophages and liver ILC1 of Ly49s KO mice were checked and no dramatic difference was found as shown in **Supplementary Fig. 1d, e**.

In Figure 1C, the expression of ly49b and ly49q is not NK specific. Ly49b is expressed in macrophages and Ly49Q is expressed on plasmacytoid dendritic cells (which likely explains why no transcript is detected for Ly49q). Authors do not examine or even discuss the implications of non-NK expression of Ly49s. Quantitative PCR of Ly49b, should be from macrophage mRNA and Ly49q from pDC mRNA (or total splenic mRNA) and should be included as a separate panel. Furthermore, Figure 1C requires proper positive and negative controls that works for both wild-type and Ly49-KO.

A: As advised, total splenic mRNA was used to detect the disruption of *Klra2* and *klra17* in Ly49-deficient mice. Meanwhile, *Nkr-p1c* and *Klrk1* were chosen as positive control (**Supplementary Fig.**

1c).

3. Figure 1D. Flow cytometry that measures Ly49Q on pDCs is needed.

A: Because Ly49Q is not expressed on NK cells, we do not think that Ly49Q deletion or not affect NK cell education. We tried to get Ly49Q antibody but it was commercially unavailable for us. To prove the absence of Ly49Q in our Ly49-deficient mice, we showed Ly49Q deletion in genomic DNA level (**Supplementary Fig.1b**). The corresponding mRNA of Ly49Q was also not detectable by RT-PCR in the splenic mRNA of Ly49-deficient mice (**Supplementary Fig. 1c, right panel**).

4. Use of name NK1.1 throughout manuscript is no longer acceptable. NKRP1C is the current acceptable name.

A: NK1.1 was all replaced by NKR-P1C.

5. Figure 2B requires proper littermate controls and should include quantification of liver NK cell population as well.

A: In our lab, we routinely use littermates or cohousing mice as controls. The quantification of liver NK cell was added in **Supplementary Fig. 1e**.

6. Authors need to address whether surface expression changes in NKG2D represent a functional consequence of Ly49 mutation or a genetic consequence caused by the significant chromosomal disruption that neighbors the NKG2D gene.

A: For this concern, we can only rule out the second possibility experimentally. We analyzed the coding sequence of NKG2D (encoded by *Klrk1*) from Ly49s KO mice, and no mutations or abnormal chromosomal relocation were found. In addition, mRNA level of *Klrk1* was comparable between WT and Ly49s KO mice (**Supplementary Fig. 1c**). Therefore, we tend to think that surface expression changes in NKG2D represent a functional consequence of Ly49 loss, as the reviewer proposed.

7. Authors need to examine why if NK development is largely intact as shown in Figure 2D, why is NKG2D affected?

A: We think that the change of NKG2D level does not mean that NK cell development is affected. Whether NKG2D is required for NK cell development is very confused ⁶ probably due to the discrepancy of genetic background. To clarify the issue, we made NKG2D-deficient mice on B6 background and found that NK cell development and differentiation were intact. These unpublished data are submitted to Journal of Leukocyte Biology and can be provided if necessary.

8. Figure 2F should include measurements of T cell and B cell repopulation as a control to show that they are not affected.

A: In **Fig. 1i**, T cells were presented.

9. Figure 3. NK cytotoxicity assay is needed as CD107a is not a sufficient surrogate for killing. NK cytotoxicity assay can be an in vitro assay and does not need to be radiometric. If authors insist that in vivo demonstration of killing is sufficient then they need to perform NK depletion in these mice to show a loss of killing.

A: As our previous study presented⁷, PK136-treated WT mice failed to eliminate β 2m-deficient target cells in **Fig. 4e**.

10. Figure 5. Authors need to verify that Ly49H is functionally deleted. Since Ly49H interacts with MCMV protein m157, authors can demonstrate this loss of function by assaying the NK response to MCMV infection.

A: B16 cells expressing m157, an active ligand for Ly49H, were generated as target cells. We found that the ectopic expression of m157 could trigger the activation of wild type, but not Ly49D/H dKO, NK cells, suggesting a functional defect in Ly49H (**Supplementary Fig. 3c**).

11. Figure 5F. One would anticipate Ly49A levels to increase under these conditions. Authors should comment on this observation.

A: We had expected that Ly49A might upregulate as a compensation of the deficiency of Ly49C and Ly49I. However, the percentage of Ly49A⁺ NK cells was decreased (**supplementary Fig.4e**). Because the stochastic expression of Ly49 receptors is under control of alternate upstream promoter, named Pro1⁸, we speculate that the reduced expression of Ly49A may be caused by the reduced transcriptional activity caused by the deficiency of Ly49C and Ly49I.

12. Figure 5F. Authors need to show Ly49C loss by either use of anti-Ly49C/I 5E6 antibody and measuring ly49C mRNA levels.

A: The flow cytometric results are shown in **Supplementary Fig. 4a**.

13. Figure 5I/J. These findings are not novel and therefore if used, it should be presented as supplementary data and not presented with the main body of results.

A: The results are presented as **Supplementary Fig. 5**.

14. Figure 5K. Missing statistical measure for comparisons between other groups.

A: Statistical analysis were performed.

15. Figure 6B. In the ly49C/I single knockout and NKG2A single knockout represent a great opportunity to test Ly49C/I and NKG2A roles in education. NKG2A educated mice should be have higher Ly49C/I (most NK cells are Ly49C/I positive) and vice versa. The authors need to perform an in vivo rejection assay for the NKG2A single knockout mice.

A: The in vivo rejection assay in NKG2A KO mice are added (**Fig. 6f, g**)

16. Figure 6I. The authors need to explain why they targeted Ly49G.

A: We initially tried to delete Ly49C and Ly49I under NKG2A KO mice, but Ly49G deletion was an off-target. Because Ly49G do not recognize the MHC-I molecules of B6 mice⁹ and play dispensable role in NK cell activity in B6 mice (**Supplementary Fig. 8c**), we kept the line for the following experiments.

17. Figure 6F. The authors need to assess development (CD11b/27) for the total knockout generated (Ly49C/G/I and NKG2A)

A: Data are shown in **Supplementary Fig. 8d, e**.

1. Corral, L., Hanke, T., Vance, R.E., Cado, D. & Raulet, D.H. NK cell expression of the killer cell lectin-like receptor G1 (KLRG1), the mouse homolog of MAFA, is modulated by MHC class I molecules. *Eur J Immunol* **30**, 920-930 (2000).
2. Oberg, L. *et al.* Loss or mismatch of MHC class I is sufficient to trigger NK cell-mediated rejection of resting lymphocytes in vivo - role of KARAP/DAP12-dependent and - independent pathways. *Eur J Immunol* **34**, 1646-1653 (2004).
3. Perarnau, B. *et al.* Single H2Kb, H2Db and double H2KbDb knockout mice: peripheral CD8+ T cell repertoire and anti-lymphocytic choriomeningitis virus cytolytic responses. *Eur J Immunol* **29**, 1243-1252 (1999).
4. Pascolo, S. *et al.* HLA-A2.1-restricted education and cytolytic activity of CD8(+) T lymphocytes from beta2 microglobulin (beta2m) HLA-A2.1 monochain transgenic H-2Db beta2m double knockout mice. *J Exp Med* **185**, 2043-2051 (1997).
5. Orr, M.T. *et al.* Development and function of CD94-deficient natural killer cells. *PLoS One* **5**, e15184 (2010).
6. Zafirova, B. *et al.* Altered NK cell development and enhanced NK cell-mediated resistance to mouse cytomegalovirus in NKG2D-deficient mice. *Immunity* **31**, 270-282 (2009).
7. Chen, S. *et al.* The Self-Specific Activation Receptor SLAM Family Is Critical for NK Cell Education. *Immunity* **45**, 292-304 (2016).
8. Gays, F., Taha, S. & Brooks, C.G. The distal upstream promoter in Ly49 genes, Pro1, is active in mature NK cells and T cells, does not require TATA boxes, and displays enhancer activity. *J Immunol* **194**, 6068-6081 (2015).
9. Ahmed, R. & Honjo, T. *Negative co-receptors and ligands*. Springer: Heidelberg New York, 2011.

REVIEWERS' COMMENTS:

Reviewer #1 (Remarks to the Author):

The authors properly addressed the points that I raised in my first review.

Reviewer #2 (Remarks to the Author):

The authors have done reasonable efforts to improve their manuscript based on the issues raised by the first version of their report. Yet, a major point remains to be modified. It is obvious from figures 6 and 7 that the role of NKG2A in NK cell education is much less profound than the role of Ly49 molecules that accounts for most of the effect of MHC Class I on the acquisition of NK cell effector functions. Therefore, the title, the abstract and the discussion should be modified accordingly.

Reviewer #3 (Remarks to the Author):

The original manuscript submitted by Zhang et al. presented a body of work describing the generation and characterization of an impressive collection of receptor mutants of the Ly49 family in mice. This work helps solidify the function of Ly49 family in NK cell development and education. While the Ly49s knockout mice and the combined NKG2A Ly49 C/G/I knockout mouse models are intrinsically valuable tools to investigate these receptors in NK biology, the experimental evidence and controls needed to effectively characterize these models in the original manuscript submission were deficient. Additionally, the manuscript previously lacked the requisite detail in the interpretation of their work against the relevant body of existing literature on Ly49 receptors in NK biology. However, in the current submission, the authors present a comprehensive and satisfactory response for each of the previous concerns stated above. Likewise, the authors have prepared to the best of their ability the requisite experimental evidence needed to substantiate their claims. The changes made have greatly strengthened this manuscript. For these reasons, I am fully satisfied that the revised manuscript prepared by the authors is both well-suited for publication in Nature Communications and also represents an important and impactful contribution to the field.